# The Impact of Spiritual Leadership on Knowledge-Hiding Behavior: Professional Commitment as the Underlying Mechanism

Yaseen Ullah *, Shahid Jan and Hamid Ullah 

Department of Management Sciences, Islamia College Peshawar, Peshawar 25120, Pakistan;
shahidjan@icp.edu.pk (S.J.); hamidullah@icp.edu.pk (H.U.)
* Correspondence: yaseenpaf@gmail.com

**Abstract:** Purpose—The purpose of this study is to investigate the impact of spiritual leadership on knowledge-hiding behavior in agriculture research institutes of Khyber Paktunkhwa, Pakistan. The study aims to analyze theoretical and empirical evidence regarding the mediation pathway, specifically professional commitment, in order to clarify the significant association between spiritual leadership and subordinates' knowledge-hiding behavior. Design/methodology—This survey-based study used cross-sectional data and a five-point Likert scale to investigate the given hypotheses. In order to address the primacy effect and mitigate any potential for common method bias, data were collected at two distinct time points, with a four-week interval between them. Smart PLS4 was used to assess a sample of 298 complete and valid responses for hypothesis testing. Findings—The results show that spiritual leadership has a negative impact on employees' knowledge-hiding behavior. Additionally, this relationship is mediated by professional commitment. Originality/value—First, in contrast to the majority of previous studies, which focused on the factors influencing knowledge sharing, the present study investigates the influence of spiritual leadership on employees' knowledge-hiding behaviors, which are two contrasting concepts. Secondly, the study empirically examined the mediation effect of professional commitment. These three variables have not previously been studied together.

**Keywords:** spiritual leadership; knowledge-hiding behavior; professional commitment; agriculture research institutes

## 1. Introduction

*Background of the Study*

Pakistan is primarily an agricultural country. Since 2013, Pakistan has become a net food importer, which led to an additional burden of USD 4.261 billion in the first nine months of fiscal year 2019 (PES 2018–2019). Furthermore, the contribution of agriculture to GDP has been decreasing over time. In comparison to the output levels of advanced countries worldwide, Pakistan's average crop yield is extremely low. Agricultural research is necessary to address the sector's underdevelopment by redirecting the focus of agricultural research towards modern approaches that promote integrated, sustainable, and profitable farming. This will lead to enhanced productivity, improved nutrition, a healthier environment, and better quality of life for people. Innovations through research and development are only possible through the effective use of knowledge. Knowledge hiding, as a prevalent workplace problem, results in significant financial losses for businesses [1]. Despite the well-established requirement for knowledge sharing, [2] found that knowledge hiding is common in many service firms, which hinders knowledge transfer [2]. Among various factors, knowledge hiding among research professionals may be the cause of the subpar performance of agricultural research institutes in Pakistan. Knowledge hiding is influenced by a wide range of contextual factors, including organizational policies, compensation systems, leadership, structure, and culture, among others. This study of [2] further

advocated additional research to investigate and comprehend the causes and effects of knowledge-hiding behaviors.

Knowledge management in contemporary organizations is a crucial and vital resource for gaining a competitive advantage and achieving success [3]. Although employees possess the knowledge [4], the ability to share it is the central concern for the success of knowledge management in contemporary organizations. Organizations frequently invest significant time and resources in acquiring new knowledge in order to maintain their competitive advantage. However, there is still a reluctance to share knowledge among employees, and they continue to be hesitant about sharing knowledge with their coworkers [5]. When critical knowledge or information is kept hidden, it can have serious consequences for organizations, ranging from negative impacts on employees [6] to project-level concerns and larger organizational inefficiencies [7]. The study of [8] Found that knowledge hiding costs Fortune 500 organizations approximately USD 31.5 billion annually. Furthermore, in 2006, the Globe and Mail surveyed more than 1,700 regular readers. Their study showed that roughly 76 percent of workers were engaged in knowledge-hiding behavior [9]. In his study, [10] discovered that 46% of respondents admitted to having engaged in knowledge-hiding behavior at least once. It is, therefore, critical to discover the factors that cause individuals, particularly research professionals, to hide their knowledge so that firms can design effective strategies to discourage this behavior. Few studies have been conducted to determine the extent of knowledge hiding because the concept of hiding knowledge is still evolving [11].

The term knowledge hiding is a three-dimensional phenomenon, namely, evasive knowledge hiding, playing dumb, and rationalized knowledge hiding approaches. Evasive hiding occurs when workers 'give wrong information and promise to provide complete information later on'. In playing dumb, the workers pretend that they do not have the required information as requested, and rationalized knowledge hiding occurs when workers provide a justification, blame someone else or say that they are not allowed by superiors to disclose or transfer such information. Studies in the area of knowledge hiding began two decades ago with the aim of understanding why people hide knowledge [12]. If some fear the loss of their power, others fear being evaluated [13]. Some individuals may perceive knowledge sharing as complex, while others may be waiting for the right conditions within their organization. If an employee distrusts the requestor or if the question is complicated, then the employees will engage in knowledge hiding. Furthermore, knowledge hiding occurs due to employees' fear of losing their status, career opportunities, or even their jobs [14]. Although organizations are attempting to develop strategies to incentivize individuals in sharing their expertise with their coworkers, nevertheless, because it is a deliberate activity [15], the employees cannot be pressured into sharing knowledge against their will [16]. Nonetheless, they may (and should) be urged and encouraged to do so. According to [17], scholarship in the context of knowledge hiding as a concept is in very nascent stage, and thus there is a need to explore the concept in varied contexts and in relation to other organizational constructs in order to enhance the theoretical legitimately of the construct. All the identified themes are evolving in terms of their density and becoming less central. Therefore, we suggest that knowledge hiding is attracting greater scholarly attention as an organizational construct.

Prior research on knowledge hiding has identified several predictors, such as organizational injustice [17] and leadership-related factors [18–22], for measuring knowledge-hiding behavior. According to [21], Spiritual leadership among others may mitigate employees' tendencies to hide knowledge, which could be an intriguing research topic for future scholars. Moerover, less attention has been paid to how to mitigate knowledge-hiding behavior. One significant factor that impacts a person's behavior in sharing knowledge with others at work is the social interactions between coworkers and leadership, as well as how one is treated while working. After conducting a comprehensive literature analysis on knowledge hiding [23] found that organizational values and leadership style are likely to have an impact on the adoption of these behaviors. Therefore, further investigation is

needed to explore the relationship between them. During a comprehensive analysis, [24] Believe that spiritual leadership is a unique and researchable topic and urges further study to enhance the breadth and depth of knowledge in this field.

Spiritual leadership, as a form of value-based leadership, has gained favor recently due to its ability to positively impact businesses. Spiritual leaders prioritize encouraging staff to uphold the organization's mission and values by offering assistance, expressing gratitude, and fostering a sense of community [25]. Research on spiritual leadership is gaining popularity [26]. Under spiritual leadership, employees are motivated by a transcendent vision and guided by hope and altruistic love. This intrinsic motivation leads them to cultivate positive social emotions, including care and concern for others, compassion, kindness, forgiveness, gratitude, and a willingness to help [27]. Additionally, by integrating ethical and spiritual principles with rational criteria in decision-making, spiritual leadership empowers staff to regulate their behavior and make morally superior decisions [27]. These emotions serve as the foundational elements for establishing and nurturing trust-based interpersonal relationships [28]. Employees who have strong, trust-based connections with one another tend to be less prone to knowledge hoarding.

Spiritual leadership is identified as one of the factors that can influence knowledge-hiding behavior. However, it is important to consider other variables, such as professional commitment, which can serve as a bridging variable between spiritual leadership and knowledge behavior. Moreover, [29] examined the direct association between spiritual leadership and affective professional commitment, with perceived organizational support as a mediator, and discovered a positive relationship between the two. Employees with high levels of professional commitment are less likely to withhold knowledge, as they perceive responding to coworkers' requests as their professional obligation. Therefore, even when working in a politically charged workplace, people with a high level of professional dedication are less likely to participate in information hiding practices [30]. Based on the views and opinions of the aforementioned writers, it is assumed that there is a relationship between spiritual leadership and knowledge-hiding behavior, which is mediated by professional commitment. The purpose of this study is to analyze the relationship between spiritual leadership, professional commitment, and their implications for knowledge-hiding behavior. Based on previous theoretical and research approaches, this study investigates the relationship between spiritual leadership (as an exogenous variable) and professional commitment (as an intervening variable), as well as the relationship between spiritual leadership and knowledge-hiding behavior (as an endogenous variable). This study utilizes Spiritual Leadership Theory (SLT), developed by and Social Action Theory as the foundation for the investigation. According to the spiritual leadership theory and social action theory, leaders who are able to transmit their personal values to others as intrinsic motivation can encourage individuals to become more engaged with the organization and develop an emotional connection to both the leader and the organization. The study of [30] also analyzes the effect of spiritual leadership on subordinates' knowledge-hiding behavior, utilizing social exchange theory as the theoretical foundation to support our hypothesis. Relationships between individuals depend on positive and beneficial exchanges and transactions, which is one of the fundamental principles of SET. When an employee realizes and experiences knowledge hiding, they are prone to retaliate, as stated by the norm of reciprocity [31] and this induces distrust among coworkers. This, in turn, leads to ineffective social exchange between them [32].

The present study makes significant contributions to the field of spiritual leadership and the existing literature on knowledge hiding. This study aims to empirically examine the correlation between knowledge-hiding behaviors among employees and spiritual leadership. Second, the study investigates the indirect relationship between knowledge hiding and spiritual leadership. It also examines professional commitment as a potential mediator that may explain this link. This study is unique in that it combines the examination of professional commitment as a mediator of spiritual leadership and knowledge-hiding behavior, two aspects that have rarely been investigated together. The study has important

implications for research officers, management, and research assistants at the agricultural research institutes in KPK, Pakistan. It will become clear that the phenomenon of knowledge hiding at work is currently at its peak in organizations, and has captured the attention of researchers from around the world. Decision makers must train and evaluate leaders and supervisors to adopt the values and practices of value-based leadership. This will help motivate employees to share critical knowledge and benefit their organizations.

## 2. Theory and Hypothesis Development

### 2.1. Spiritual Leadership and Knowledge-Hiding Behavior

In their detailed study, [33] argued that spiritual leadership is characterized by a dedication to integrity, goodness, cooperation, knowledge, thoroughness, and connectedness. Both positive and negative reciprocity norms are present within organizational contexts. Positive reciprocity refers to the inclination to react positively when treated favorably, while negative reciprocity refers to the inclination to react negatively when treated unfavorably [34]. According to [35] social learning theory, individuals in subordinate positions often perceive their superiors as role models. Consequently, they tend to imitate the behavior of their superiors based on how they are treated. Spiritual leadership is characterized by three defining traits: altruistic love, faith/hope, and vision. These traits serve to propel and inherently encourage subordinates to seek meaning in their work. According to [36], individuals who follow these leaders tend to engage in pro-social behaviors and demonstrate genuine concern for their colleagues, thereby enhancing the quality of their interpersonal relationships.

An essential element of spiritual leadership is altruistic love, which can be defined as "a sense of wholeness, harmony, and well-being achieved through caring, concern, and admiration for both oneself and others" Spiritual leaders demonstrate altruistic love through their actions and behaviors, such as caring for others, expressing gratitude, showing compassion, and practicing kindness. Spiritual leadership embraces social and spiritual ideals, such as honesty, fairness, and ethical conduct, while making decisions and interacting with others [37]. Positive behaviors can be passed on from one person to the next, according to the conservation of resource theory [38]. According to [39], organizations hinder their own growth, learning, and adaptation to the ambiguous business environment by concealing knowledge and expertise that could lead to improvement. Findings of [39] after examining the association between workplace spirituality and the dimensions of knowledge-hiding behavior showed that workplace spirituality reduced evasive knowledge-hiding behavior. Furthermore, workplace spirituality diminished playing dumb knowledge-hiding behavior; however, it did not diminish rationalized knowledge-hiding behavior, as the relationship appears to be insignificant.

This study suggests that subordinates can adopt altruistic tendencies, a sense of self-transcendence, and spiritual values. Consequently, these subordinates would demonstrate the same behaviors of compassion and concern for others, fulfilling their spiritual needs. As a result, employees who work with spiritual leaders would be less inclined to engage in socially unproductive practices such as knowledge hiding. Instead, they would model spiritual ideals and selfless love to their coworkers in order to improve their relationships and reduce knowledge-hiding behavior [39]. Employees who have internalized the company vision and incorporated it into their personal values perceive sharing professional information as a desirable achievement [40]. As a result, we conclude that spiritual leadership minimizes employees' knowledge-hiding behavior. The current study proposes the following hypothesis based on these theoretical grounds.

**H1:** *There is a significant and negative relationship between spiritual leadership and knowledge-hiding behavior.*

## 2.2. Spiritual Leadership and Professional Commitment

Spiritual leadership is a management style that emphasizes the importance of love, hope, and vision in motivating subordinates [41]. Leaders who have feelings of love will make people feel appreciated, respected, and valued. This, in turn, can intrinsically motivate individuals to adopt the leader's characteristics and values. According to social action theory and spiritual leadership theory, when leaders are able to effectively communicate their personal ideals to followers, this can serve as intrinsic motivation. As a result, individuals are more likely to feel a strong connection to organizations with high engagement and strive to build an emotional relationship with the leader. This results in a strong organizational and professional commitment [42]. This is consistent with the finding by [43] that employees experience a higher level of affective professional commitment when there is a greater presence of spiritual leadership.

The aim of spiritual leadership is to tap into the fundamental needs of both leaders and followers for spiritual well-being. This is achieved through fostering a sense of calling and membership, which in turn promotes a shared vision and values among individuals, empowered teams, and the organization as a whole. Ultimately, this approach aims to enhance the well-being and professional commitment of employees. Spiritual leadership research combined with professional commitment is still uncommon. According to [43], the more visible the spiritual leadership, the greater the effective professional commitment of employees. Spiritual leadership theory and social action theory state that when leaders are able to transmit their personal values to others as intrinsic motivation, individuals will be more inclined to align themselves with organizations that encourage active participation and develop an emotional connection with the leader. This is in line with [43], who found that the more visible the spiritual leadership, the greater the commitment felt by employees. Thus, the following hypothesis is proposed:

**H2:** *There is a significant relationship between spiritual leadership and professional commitment.*

## 2.3. Professional Commitment and Knowledge-Hiding Behavior

One's dedication to a specific career is known as professional [44]. Professional commitment refers to having a firm belief in the objectives and core principles of a profession, making significant efforts on behalf of the profession, and having a desire to become a part of it [45]. According to a study on accountants' professional dedication [46], employees who are professionally committed put in effort for the benefit of the profession. As a result, they internalize the profession's success or failure as their own success or failure. Additionally, there are many different types of professional commitment, including professional affective commitment (desire-based), professional continuance commitment (cost-based), and professional normative commitment (obligation-based) [47,48].

A correlation between professional commitment and knowledge hiding among students was examined by [49]. The results of this study showed that learners with a high level of professional commitment perceived their research subject more emotionally than intellectually. They also viewed their requesters as potential collaborators rather than competitors. As a result, individuals who have a strong internal commitment to their profession may openly resist the act of concealing knowledge, whereas those with a lower level of professional commitment may engage in knowledge hiding when specifically requested to do so. According to [50], when managers receive bonuses and incentives, their professional commitment to the company increases, which ultimately leads to a reduction in knowledge-hiding behavior. Hence, the study seeks to examine the relationship between employees' professional commitment and their knowledge-hiding behavior. Thus, the following hypothesis is proposed:

**H3:** *There is a significant relationship between professional commitment and knowledge-hiding behavior.*

*2.4. The Mediating Role of Professional Commitment between Spiritual Leadership and Knowledge-Hiding Behavior*

Indeed, by cultivating positive social emotions such as compassion, concern for others, and gratitude through altruistic love, spiritual leadership inspires subordinates to assist and prioritize the well-being of their colleagues [51]. Through an interactional method (a bottom-up strategy that promotes two-way communication) [52], spiritual leaders convey a compelling transcendent vision to employees, increasing employee involvement in critical decisions and removing uncertainties related to their professional roles. Additionally, spiritual leaders, through their visionary perspective, inspire and motivate their followers to assist others, foster meaningful social connections, and engage in actions that positively influence the relational well-being of their colleagues. This instills a sense of meaning and purpose in the lives of those followers [53]. As a result, spiritual leadership can help employees develop strong relationships with their coworkers, leading to mutual support and a positive feedback loop. They also view the sharing of professional knowledge as a personally rewarding achievement. According to social learning theory, subordinates generally view their superiors as role models, and as a result of how they are treated, they also adopt those behaviors. Spiritual leaders, as stated by [54], encourage important individual and organizational behaviors, such as promoting knowledge sharing and discouraging knowledge hiding.

The social action theory and spiritual leadership theory serve as the theoretical foundations for using professional commitment as a mediator. The values of spiritual leadership are conveyed to subordinates, motivating them in their profession. Employees working under spiritual leadership would not squander resources, as it may weaken their devotion to their profession and result in the loss of other valuable assets, such as knowledge. Instead, individuals are more inclined to engage in constructive behaviors such as altruism and pro-sociality, which can help them, acquire future resources. In light of prior research, we believe that spiritual leadership improves employee commitment to their professions, which has a negative impact on knowledge-hiding behavior [55].

The current study asserts that spiritual leadership strengthens employees' professional commitment through altruistic love, hope/faith, and transcendent vision. This, in turn, helps prevent or reduce knowledge-hiding behavior, as leaders and their subordinates serve as significant sources of commitment. The] notion of organizational commitment is similar to the definition of professional commitment. Professional commitment has three dimensions: emotive, ongoing, and normative [56]. These theoretical considerations support the proposed hypothesis of this study, which suggests that professional commitment acts as a mediator between spiritual leadership and knowledge-hiding behaviors. It is necessary to conduct further research. Social exchange theory posits that individuals are more likely to develop long-term relationships and form emotional attachments to leadership when the organization fosters mutual support and collaboration between individuals and leaders. Emotional attachments are formed when individuals receive values that align with both the values of leaders and their own individual values. Therefore, when conformity is valued, it will increase the employee's commitment. Based on spiritual leadership theory, leaders who show love enhance employees' intrinsic motivation. Therefore, employees will be interested in adopting the values that the leader embodies as this will enhance their professional commitment. Thus, we propose:

**H4.** *Professional commitment mediates the relationship between spiritual leadership and knowledge-hiding behavior.*

## 3. Methodology

*3.1. Sample and Data Collection*

This study used a questionnaire-based survey approach. Participants in the study were research professionals working in agricultural research institutes in KPK, Pakistan. The selected population for this study was chosen because Pakistan is an agricultural country,

and research institutes focused on agriculture are essential due to the sector's significance to the country's economy and its role in job creation. As a result, we consider the current study that examines employees' knowledge-hiding behaviors at agriculture research institutes in Khyber Pakhtunkhwa, Pakistan, to be important. The study included a population size of 1230 research professionals. According to Ume Sekaran, a sample size of 291 is sufficient for generalizability of the findings for a population of 1200. Females accounted 51 (17.11%) members of the sample, while males made up 247 (82.89%) members. The respondents ranged in age from 24 to 54 years. The average tenure in an organization was 5.8 years. All of the respondents had a bachelor's degree or higher. The sample included a variety of positions, such as research officers, research assistants, and HR managers. The purposive sampling technique was used to collect data from professionals in agricultural research. A total of 298 usable responses were obtained from various research institutes in the Khyber Pakhtunkhwa region of Pakistan.

Questionnaires were distributed both in person and via email to each institute. Data were gathered twice, with a four-week interval between each collection. This is consistent with the recommendation of [57] to mitigate the risk of common method bias. This study utilizes a two-wave method. First, the researchers contacted and briefed the HR managers of the organization about the study and the purpose of data collection. The interested parties were given instructions for completing the survey and were assured of the confidentiality of their provided information. At time 1, the respondents provided basic demographic data about themselves and completed a five-point Likert scale to score the mediating variable, professional commitment, and the predicting variable, spiritual leadership. At Time Point 1, we received 341 responses. We contacted the same group of respondents again four weeks later to gather their feedback on their knowledge-hiding behavior. A total of 298 responses were used after eliminating incomplete and disinterested responses.

### *3.2. Measures*

In this study, spiritual leadership was considered as the independent variable, knowledge hiding was considered as the dependent variable, and professional commitment was considered as the mediator. A total of 39 items were used on a 5-point Likert scale. Respondents can utilize the Likert scale to more accurately assess the intensity of their emotions or behaviors. This allows for a more precise evaluation of the variables being studied, which may be challenging to measure directly [58]. Another advantage of using this scale in the questionnaire is the ease of administration and the low cost of collecting data from a large number of individuals. On these grounds, the study utilized a five-point Likert scale to gather data. The variables of the suggested model are based on reliable and valid measurements that have been utilized by other researchers in previous studies and have strong psychometric properties. Spiritual leadership was measured using an 18-item scale developed by [59]. Moreover, a 12-item scale was adopted to measure knowledge hiding, which consists of three dimensions. Similarly, to measure professional commitment, we adopted the 9-item scale used by [60]. On this scale, respondents rated the extent to which they agreed or disagreed with each statement.

### 4. Data Analysis and Results

This study employed PLS-SEM for data analysis. The preference for PLS-SEM is based on several reasons. Firstly, it is suitable for evaluating complex models that contain several auxiliary variables, such as mediating and moderating variables [61]. Additionally, PLS-SEM is not constrained by the assumption of normality. Finally, PLS-SEM is preferred when there is a small sample size (as in this study). Utilizing SPSS and AMOS, the study first tested for common method bias. The latest version of the smart PLS 4.0 was used to analyze both the measurement and structural models. The data analysis was performed in two stages: (i) analysis of the measurement model, and (ii) analysis of the structural model. The measurement model includes the measurement of the indicators, assessing their reliability and validity, as well as evaluating the reliability and validity of the formative

constructs. The investigation of the relationships between latent variables or constructs is conducted using the structural model. The analysis includes the path coefficients, direct and indirect effects, mediating effects, as well as the coefficient of determination (R2).

### 4.1. Common Method Bias

The majority of cross-sectional studies that use the same approach to collect data from a single source are vulnerable to common method bias (CMB). The current study addressed this matter by collecting and analyzing data using the two-wave data collection method. We performed a collinearity test by measuring the "VIF" values, and we discovered that all of the values are below the cutoff value of 3.3. The model demonstrates that there is no risk of common method bias.

### 4.2. Measurement Model

The reliability, convergent, and discriminant validity of all of the first-order latent variables were evaluated in the measurement model. Each study variable's reliability was evaluated at both the item and construct levels. All variables' Cronbach's alpha (CA) values were above the cutoff of 0.70 (Table 1). The internal consistency of all the constructs is demonstrated by the fact that the coefficient alpha (Cronbach's alpha) for all the dependent, independent, and mediating variables ranged between 0.961 and 0.993, which is above the required level of 0.70. [62]. Their composite reliability (CR) also indicates good construct reliability. The "AVE" values for the three constructs were computed to evaluate the convergent validity of the variables. It was discovered that the values ranged from 0.875 to 0.981, all of which were above 0.5 (Table 1). This indicates that all three constructs have a high level of convergent validity.

**Table 1.** Construct reliability and validity.

| Constructs | No of Items | CR | Cronbach's Alpha | AVE | Mean | SD |
|---|---|---|---|---|---|---|
| KH | 12 | 0.961 | 0.940 | 0.981 | 9.570 | 5.240 |
| PC | 09 | 0.984 | 0.982 | 0.875 | 9.500 | 3.230 |
| SL | 18 | 0.993 | 0.993 | 0.892 | 10.200 | 5.250 |

Note. CR: composite reliability; AVE: average variance extracted; SD: standard deviation; KH: knowledge hiding; PC: professional commitment; SL: spiritual leadership.

### 4.3. Discriminant Validity

Discriminant validity, in contrast to convergent validity, assesses the extent to which one latent construct is distinct from another [63]. Through AVE, discriminant validity is determined [64]. The Fornell–Larcker test was employed in the study to determine whether the square root of the construct's AVE exceeds its largest correlation with other research constructs or not (Table 2). Additionally, the findings show that all of the latent variables have correlation values that are lower than the square roots of the AVE. The constructs have been confirmed to be unique and meet the requirements for discriminant validity.

**Table 2.** Discriminant validity.

|  | KH | PC | SL |
|---|---|---|---|
| KH | 0.944 | | |
| PC | −0.279 | 0.936 | |
| SL | −0.197 | 0.824 | 0.945 |

### 4.4. Outer Loadings

Second, it was examined whether the outer loadings (Table 3) of each indicator were stronger on the target construct. To evaluate the reliability of the indicators, we examined the item-to-construct loadings and found that all values exceeded the cutoff point of 0.70. Our findings validate the suitability of the measures, indicating that each item effectively represents its corresponding latent variable.

**Table 3.** Outer loadings.

|  | KH | PC | SL |
|---|---|---|---|
| EKH | 0.935 | | |
| PD | 0.934 | | |
| RKH | 0.962 | | |
| PC1 | | 0.963 | |
| PC2 | | 0.944 | |
| PC3 | | 0.968 | |
| PC4 | | 0.893 | |
| PC5 | | 0.945 | |
| PC6 | | 0.893 | |
| PC7 | | 0.939 | |
| PC8 | | 0.958 | |
| PC9 | | 0.914 | |
| SL1 | | | 0.977 |
| SL2 | | | 0.968 |
| SL3 | | | 0.977 |
| SL4 | | | 0.966 |
| SL5 | | | 0.952 |
| SL6 | | | 0.949 |
| SL7 | | | 0.938 |
| SL8 | | | 0.969 |
| SL9 | | | 0.966 |
| SL10 | | | 0.961 |
| SL11 | | | 0.943 |
| SL12 | | | 0.859 |
| SL13 | | | 0.924 |
| SL14 | | | 0.893 |
| SL15 | | | 0.848 |
| SL16 | | | 0.966 |
| SL17 | | | 0.965 |
| SL18 | | | 0.966 |

### 4.5. HTMT Ratio

Third, when assessing discriminant validity, we examined the ratio of correlations, specifically the HTMT values. This method outperforms both the Fornell–Larcker criterion and the evaluation of cross-loadings [65]. All the scores of HTMT (Table 4) are below 0.85, indicating that the current study has achieved discriminant validity [66].

**Table 4.** HTMT Ratio.

| Constructs | KH | PC | SL |
|---|---|---|---|
| KH | | | |
| PC | 0.273 | | |
| SL | 0.202 | 0.829 | |

Note. KH: knowledge hiding; PC: professional commitment; SL: spiritual leadership.

### 4.6. Coefficient of Determination (R2)

The most important factor to consider when evaluating the structural path model for the dependent variable is the determination of R2. The R2 value indicates how well the dependent variable can explain the total variance of the independent factors [67]. However, it is commonly believed that the greater the number of variants, the higher the R2 value. R2 values are now more acceptable due to the aforementioned categorization and the utilization of the PLSSEM technique as the primary statistical analysis in the current study. KH has a value of 0.810, and PC has a value of 0.667 for R2. Table 5 measures the R2 value of the constructs.

**Table 5.** Coefficient of determination (R2).

|  | R-Square | R-Square Adjusted |
| --- | --- | --- |
| KH | 0.815 | 0.810 |
| PC | 0.678 | 0.677 |

### 4.7. Structural Model

The structural model, which illustrates the direct and indirect effects of spiritual leadership on knowledge hiding with professional commitment as a mediating variable, is depicted in Figure 1. Similarly, Figure 1 also presents the path coefficients. Instead of using goodness-of-fit (GoF) metrics, as is the case with covariance structure analysis, PLS-SEM checks the overall structural model fit using the following criteria: VIF values, significance and direction of the relationships, R2 and "f2" (effect size) values and SRMR values. Calculating the VIF values of the focal constructs revealed that there was no threat from collinearity because they were less than 5.0 (Table 6).

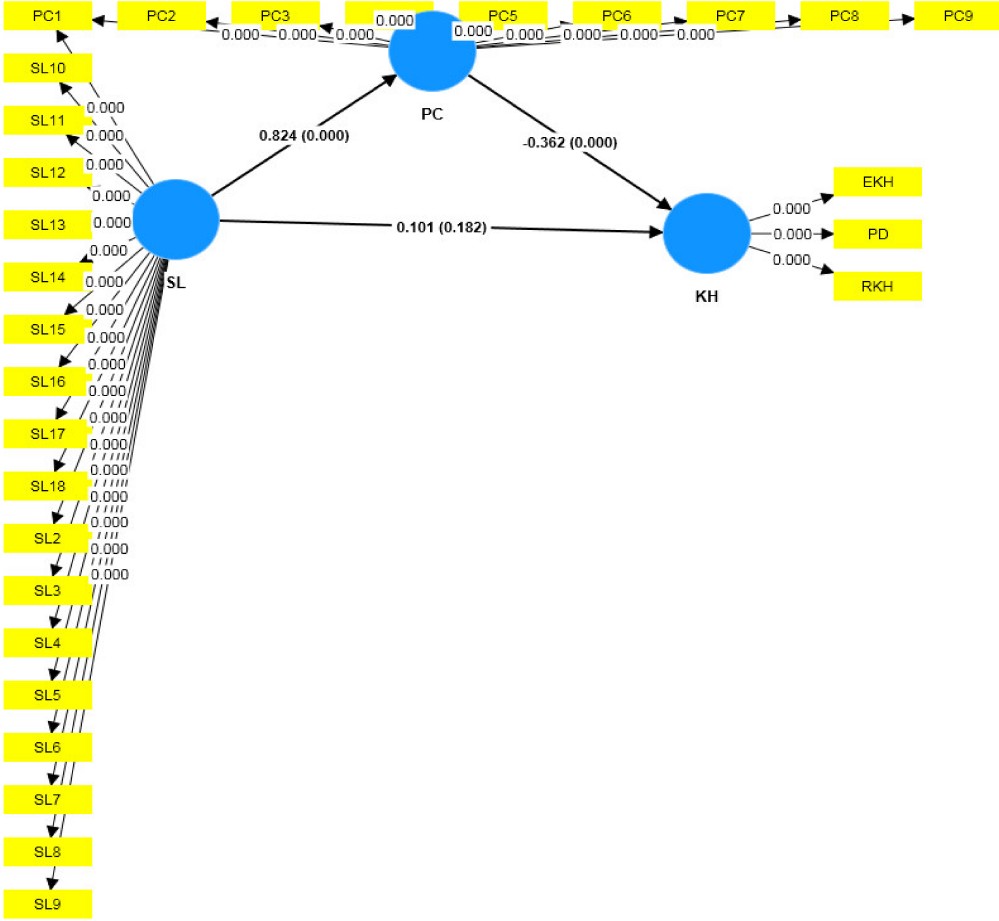

**Figure 1.** Structural model with path analysis.

**Table 6.** Collinearity statistics of structural mode l (inner VIFs).

|  | KH | PC | SL |
|---|---|---|---|
| KH |  |  |  |
| PC | 3.11 |  |  |
| SL | 3.11 | 1.00 |  |

Note. KH: knowledge hiding; PC: professional commitment; SL: spiritual leadership.

In the PLS-SEM bootstrapping process, 5000 samples were used to determine the relationship between spiritual leadership, knowledge hiding, and the mediator professional commitment. Table 7 displays the results for exogenous and endogenous variables, including their Beta coefficients, t-values, $p$-values, and decisions based on the results. Table 4 displays all of the results for direct hypotheses. Since there are strong relationships between spiritual leadership and knowledge hiding, spiritual leadership and professional commitment, and professional commitment and knowledge hiding. Hence, all three direct hypotheses (H1, H2, and H3) are accepted. Briefly stated, the study's findings reveal significant associations as follows: (i) SL -> KH (β = −0.197, $p$ = 0.000, and t = 5.489); (ii) SL -> PC (β = 0.824, $p$ = 0.000, and t = 4.412); and (iii) PC -> KH (β = −0.362, $p$ = 0.000, and t = 25.403).

**Table 7.** Total effects.

|  | Original Sample | Standard Deviation | T Statistics | $p$ Values | Decision |
|---|---|---|---|---|---|
| SL -> KH | −0.197 | 0.045 | 4.412 | 0.000 | Accepted |
| SL -> PC | 0.824 | 0.032 | 25.403 | 0.000 | Accepted |
| PC -> KH | −0.362 | 0.066 | 5.489 | 0.000 | Accepted |

These are direct relationships which are significant.

*4.8. Mediation Analysis*

PLS-SEM and bootstrapping are used to measure this model in accordance with the structural path model. A total of 5000 samples are utilized to assess the mediating role of professional commitment between spiritual leadership and knowledge-hiding behavior. According to [61], bootstrapping is suitable for mediation analysis in PLS-SEM. Employing the bootstrapping approach of [66], PLS-SEM is used to measure the mediating effect. This study examines the indirect or mediating effect of professional commitment on the relationship between spiritual leadership and knowledge-hiding behavior. Further results are shown in Table 8: SL -> PC -> KH (β = −0.302, $p$ = 0.000, and t = 5.028). Based on these results, the mediating effect is significant; therefore, H4 is accepted.

**Table 8.** Mediation results.

| Total Effect (SL→KH) | | | Direct Effect (SL→KH) | | | | Indirect Effect of SL on KH | | | | | |
|---|---|---|---|---|---|---|---|---|---|---|---|---|
| Coefficient | *t*-Value | *p*-Value | Coefficient | *t*-Value | *p*-Value | Hypothesis | Coefficient | SE | *t*-Value | *p*-Value | Percentile Bootstrap 95% Confidence Interval | |
| | | | | | | | | | | | LCI | UCI |
| −0.199 | 4.412 | 0.000 | 0.103 | 1.334 | 0.182 | SL→PC→KH | −0.302 | 0.059 | 5.028 | 0.000 | −0.428 | −0.106 |

Note. SE: standard error; SL: spiritual leadership; PC: professional commitment; KH: knowledge hiding.

The structural model's effect size (f2) is shown in Table 5, and when matched with [67] rules of small (0.02), medium (0.15), and large (0.35), the results in Table 9 show that the effect size of spiritual leadership on subordinate's knowledge hiding is 0.04, which shows a small effect size. The effect size of spiritual leadership on professional commitment is 2.11, indicating a large effect size; whereas the effect size of professional commitment on knowledge hiding is 0.046, indicating a large impact size.

**Table 9.** Effect size (f2).

| | KH | PC | SL |
|---|---|---|---|
| KH | | | |
| PC | 0.046 | | |
| SL | 0.040 | 2.11 | |

Note. KH: knowledge hiding; PC: professional commitment; SL: spiritual leadership.

Lastly, the value of SRMR is 0.06, which is below the cutoff level of 0.10, confirming the PLS structural model's overall fit.

## 5. Conclusions

### 5.1. Discussion and Conclusion

The present study examined the correlation between spiritual leadership and knowledge-hoarding behavior among employees, as well as the mediating influence of professional commitment. The first hypothesis (H1) examines the empirical investigation of the relationship between spiritual leadership and subordinate's knowledge-hiding behavior. The results showed that the β-value was negative at −0.197, and the t-statistic was above the threshold level, i.e., 4.412 > 1.96, with a *p*-value of 0.000. Therefore, H1 was accepted and found to have a significant negative effect. Spiritual leadership and workplace spirituality share the same values, making them essentially the same concept. The findings of this study are in line with the studies conducted by [68]. Their results revealed that the dimensions of workplace spirituality, namely meaningful work and values alignment, play significant roles in reducing knowledge hiding. This reduction was observed across workplace knowledge hiding's three dimensions: evasive hiding, rationalized hiding, and "playing dumb". Both studies found a negative relationship between workplace spirituality and knowledge-hiding behavior. In agricultural research institutes in Pakistan, spiritual values such as compassion, altruistic love, and honesty play a vital role in shaping a cooperative environment for research professionals to share their valuable knowledge for the betterment of organizational success.

The second hypothesis (H2) examines the empirical evidence of the relationship between spiritual leadership and employees' professional commitment. The results showed that the β-value was found to be positive, at 0.824, and the t-statistic was observed to be 25.403, which is greater than 1.96, with a p-value of 0.000. Therefore, H2 was accepted and was found to have a positive and significant effect. The findings of this study are similar to those of [69], who found a significant relationship between spiritual leadership and employees' professional commitment. In agriculture research institutes of Pakistan, research professionals are the backbone of R&D for innovations in the field of agriculture. As employees draw inspiration from their leaders, it is necessary for leadership to practice the values of spiritual leadership and make their subordinates feel valued. This will foster a sense of love for their organization and profession. The third hypothesis (H3) empirically examined the relationship between professional commitment and knowledge-hiding behavior. The results showed that the β-value was negative at −0.362, and the t-statistics were greater than the threshold level, with a value of 5.489 > 1.96, and a *p*-value of 0.000. Hence, H3 was found to have a significant negative effect. The results are similar to the study conducted by [22], which found that high professional commitment negatively affected knowledge hiding, while low professional commitment positively

affected knowledge hiding. The fourth hypothesis (H4) examines the empirical mediating role of professional commitment between spiritual leadership and knowledge-hiding behavior. The results showed that the β-value was positive, at −0.302, and the t-statistics were above the threshold level, as 5.028 > 1.96, with a p-value of 0.000. Thus, H4 was accepted, and was found to have a significant negative effect. There is no proper evidence of this mediating relationship in the previous literature. However, these results could be related to the study conducted by [68], who reported a negative relationship between workplace spirituality and knowledge-hiding behavior. They also confirmed the mediating role of organizational identification.

The findings are further supported by SET, which suggests that when an employee perceives leadership as supportive and displaying spiritual values at work, he or she is motivated. A sense of cooperation enables them to exhibit productive workplace behavior, such as helping coworkers and sharing valuable knowledge instead of hiding it. Our findings were also supported by the conservation of resources theory, which suggests that employees who consider knowledge to be a valuable resource that they do not want to waste tend to hide it from others. The study's findings indicate that the mediator, namely professional commitment, fully mediates the negative relationship between spiritual leadership and knowledge-hiding behavior. The current study is one of the few empirical studies that have examined the link between knowledge-hiding behavior among employees and spiritual leadership. The majority of previous studies have emphasized knowledge sharing rather than knowledge hiding. Contrary to popular misconceptions, these two notions are separate constructs with distinct antecedents and different outcomes, rather than being two endpoints of the same continuum. The study also examined the indirect effect of professional commitment as a mediator to help explain the relationship between the focused constructs. The mediation process clarifies how spiritual leadership helps mitigate the maladaptive behavior of knowledge hiding in the workplace.

### 5.2. Implications and Contributions

The study's results have several theoretical and managerial implications. The first implication is that spiritual leadership is widespread in the workplace and has a positive impact on job performance and, most importantly, the profitability of the organization. Because of the interpersonal relationship between leaders and subordinates, all knowledge-intensive sectors are at substantial risk if they have a destructive culture of knowledge concealment. Agriculture research institutions rely heavily on key judgments based on real-time data or information. If, for some reason, the decision makers throughout the organizational hierarchy do not have access to this key information, the results could be disastrous. Organizations may even lose their competitive advantage as a result of internal conflict that leads to knowledge hiding.

Knowledge hoarding, as a response to dysfunctional leadership, can impede innovation and creativity in the workplace and foster a culture of secrecy, which is not aligned with the expectations of businesses and other stakeholders. It is very challenging to completely eliminate this interpersonal annoyance from the workplace [70]. Organizations can, however, adopt a zero-tolerance policy in this regard. This will reassure employees that sufficient checks and balances have been implemented by the company to guarantee fair and respectful treatment at work. Second, organizations may consider training and preparing their leadership to exhibit typical spiritual behaviors, such as altruistic love, cooperation, and support for followers. Third, organizations could also consider providing therapy and support services to their members in order to align with the spiritual values of their leadership in the workplace.

### 5.3. Limitations and Further Study

Although this study provides valuable insights into the direct and indirect impacts of spiritual leadership on employees' knowledge-hiding behaviors, it is important to acknowledge its limitations. The study is limited by its reliance on a single source and a singular

method for data collection, thereby introducing a potential bias in the findings. Although the present study adhered to the data collection and analysis guidelines, measures were implemented to mitigate the potential influence of common method bias (CMB). However, future research could potentially derive greater benefits by incorporating objective data from multiple sources in order to enhance our comprehension of this relationship. Second, the cultural perspective can significantly influence research outcomes. In order to advance knowledge in the field, it is imperative for future researchers to undertake analogous studies from diverse cultural perspectives. Longitudinal research has the potential to offer confirmatory evidence for the findings of the present study. Additionally, forthcoming research endeavors may uncover supplementary variables that contribute to the amplification of knowledge concealment within the organizational setting. Psychological ownership of knowledge can serve as a moderating factor, potentially mitigating the negative impact of knowledge hiding. We also advocate for the acknowledgment of supplementary leadership styles, such as transformational leadership, as a mitigating element of knowledge concealment tendencies. This is due to the fact that research indicates that both possess significance and resemblances in their approach to management, specifically through the utilization of a value-based leadership style.

**Author Contributions:** Conceptualization, Y.U. and H.U.; methodology, H.U.; software, H.U.; validation, Y.U., H.U. and S.J.; formal analysis, Y.U.; investigation, Y.U.; resources, Y.U.; curation, H.U.; writing—Y.U., H.U. and S.J.; visualization, Y.U.; supervision, H.U.; project administration, S.J. All authors have read and agreed to the published version of the manuscript.

**Funding:** This research received no external funding.

**Informed Consent Statement:** Informed consent was obtained from all subjects involved in the study.

**Conflicts of Interest:** The authors declare no conflict of interest.

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
