# Peer review of "The Impact of Spiritual Leadership on Knowledge-Hiding Behavior: Professional Commitment as the Underlying Mechanism"

_knowledge, doi:10.3390/knowledge3030029_

Round 1

Reviewer 1 Report

Extensive survey of relevant literature is demonstrated.

Background discussion of the hypotheses is appropriate.

Research design is clearly discussed.

Findings of the study are matched with findings from other studies.

Grammar/Spell check is required in number of places. The following are just few examples

page 2. "research professionals...." (plural)

"...has serious consequences..."

"admitted to having engaged..."

"i.e., evasive..." (also in 5. Conclusion, Line 27 i-e should be i.e.,)

"they don't have the required"

Please improve sentence: "Spiritual leadership's through their three defining..."

References 99-101 are paragraph splits of the same reference.

Author Response

Point-1. Minor editing of English language has been carried out.

Point-2. Grammar/Spell corrected.

Point-3. On page 2, plural changed into singular.

Point-4. Various sentences improved.

Point-5. References 99-101 corrected.

Reviewer 2 Report

This is a good and much needed study of knowledge-hiding in the workplace as this phenomenon is influenced by spiritual leadership and professional commitment.  The study is well-grounded in qualitative research and data and makes substantial findings available to the reader. However, there are several problems that detract from the study. First, there are several misspelled words: “prim” for “prime,” “professional” for “professionals,” “dump” for “dumb,” “hypotheses” for “hypothesis,” and, maybe “SET” for “SLT.”  In some places an article like “a,” “an,” or “the” is missing.  Therefore, grammatical editing is needed.  This article would benefit greatly from a professional English editing service. Next, the article is repetitive in places. Once the point is made it does not need to be made again except in a summary.  Next, the author(s) should clarify whether the central problem in the article involves research professionals hiding knowledge from administrators, or employees hiding knowledge from co-workers. Lastly, although “knowledge-hiding” and “knowledge-hoarding” can be used interchangeably, it is less distracting to use only one of these terms consistently throughout the whole paper.  This is a worthy study, and with these recommended revisions, it should be published.

There are several spelling and grammatical errors.  I recommend  an English editing service.

Author Response

All required crevisions has been carried out

Round 2

Reviewer 2 Report

The manuscript has been significantly improved by the revisions made.  I am grateful for the opportunity to review this study again. There are still a few grammatical errors.  For example, use of the plural term "hypotheses" when the sentence requires the singular term "hypothesis."  I am sure that these corrections will be made by the Academic Editor and typesetter.  

As I mentioned before, a few grammatical errors remain.  For example, use of the plural term "hypotheses" when the sentence requires the singular term "hypothesis."